# An Evolutionary Perspective of Dopachrome Tautomerase Enzymes in Metazoans

**DOI:** 10.3390/genes10070495

**Published:** 2019-06-28

**Authors:** Umberto Rosani, Stefania Domeneghetti, Lorenzo Maso, K. Mathias Wegner, Paola Venier

**Affiliations:** 1Department of Biology, University of Padova, Padova, 35121, Italy; 2Alfred Wegener Institute (AWI)—Helmholtz Centre for Polar and Marine Research, Wadden Sea Station Sylt, List auf Sylt25992, Germany

**Keywords:** DCT, DCE, *yellow*, dopachrome tautomerase, MRJP, oyster, melanin, *Mytilicola*

## Abstract

Melanin plays a pivotal role in the cellular processes of several metazoans. The final step of the enzymically-regulated melanin biogenesis is the conversion of dopachrome into dihydroxyindoles, a reaction catalyzed by a class of enzymes called dopachrome tautomerases. We traced *dopachrome tautomerase* (DCT) and *dopachrome converting enzyme* (DCE) genes throughout metazoans and we could show that only one class is present in most of the phyla. While DCTs are typically found in deuterostomes, DCEs are present in several protostome phyla, including arthropods and mollusks. The respective DCEs belong to the *yellow* gene family, previously reported to be taxonomically restricted to insects, bacteria and fungi. Mining genomic and transcriptomic data of metazoans, we updated the distribution of DCE/*yellow* genes, demonstrating their presence and active expression in most of the lophotrochozoan phyla as well as in copepods (Crustacea). We have traced one intronless DCE/*yellow* gene through most of the analyzed lophotrochozoan genomes and we could show that it was subjected to genomic diversification in some species, while it is conserved in other species. DCE/*yellow* was expressed in most phyla, although it showed tissue specific expression patterns. In the parasitic copepod *Mytilicola intestinalis* DCE/*yellow* even belonged to the 100 most expressed genes. Both tissue specificity and high expression suggests that diverse functions of this gene family also evolved in other phyla apart from insects.

## 1. Introduction

Innate immunity represents an early defense system in vertebrate and invertebrate animals [1]. It involves physical barriers, phagocytic cells and a variety of proteins which include recognition and effector molecules, that coordinately act to block infections and promote inflammation as a further defense mechanism [1]. One of the most ancestral innate defense is encapsulation, a process referring to the binding of multiple cells to a large invader, that cannot be phagocytized by a single cell [2,3]. This process begins soon after the recognition of the invader by cellular sensors of pathogen-associated molecular patterns (PAMPs) and it is characterized by abundant production of reactive oxygen species (ROS), thus facilitating the elimination of the intruder [3]. In arthropods, the production of a capsule surrounding the invader is followed by its melanization to enclose the invader into an environment with high concentration of ROS [3]. Accordingly, the melanin’s ancestral role was assumed to be linked to this defense mechanism [4], even if melanization is also involved in other biological processes like wound healing, clot formation, pigmentation and UV-protection [5,6].

Although melanotic encapsulation is mostly described in arthropods [7], the presence of dark, melanin-like granules surrounding pathogens has also been reported in several mollusk species. Examples for this come from the gastropod *Achatina fulica* infected by nematodes [8], *Roseovarius crassostreae* infecting *Crassostrea virginica* and *Vibrio tapetis* infecting *Ruditapes phlilippinarum* [9]. Here, all pallial epithelia of bivalves were capable of melanization [9], but most of the genes involved in the melanogenic pathway were highly expressed in the mantle edge [10,11], a tissue involved in both defense mechanisms and shell formation in bivalve and gastropod mollusks [12,13,14].

Melanin originates from tyrosine and related phenolic compounds, which are hydroxylated into *o*-diphenols, oxidized into quinones and polymerized to obtain melanin [15]. Although most of the gene complements of the melanogenic pathways have been traced in vertebrates and arthropods [16,17], the enzyme functions promoting the last enzymatic reaction (EC 5.3.3.12), namely the conversion of dopachrome into 5,6-dihydroxyindole (DHI) or 5,6-dihydroxyindole-2-carboxylic acid (DHICA) [18], were seldom investigated. This enzyme, called dopachrome tautomerase (DCT) [15], was initially discovered in mice in 1980 [19]. Nowadays, it is known that human DCT regulates the production of ROS facilitating human papillomavirus infection [20], it is involved in the progression of melanomas [21] and it protects melanocytic cells from ROS damage [22]. Subsequent to DCT, an enzyme with an equivalent function was characterized in the insect *Manduca sexta* [23]. This gene, known as dopachrome decarboxylase/tautomerase or dopachrome converting enzyme (DCE), was previously discovered in *Drosophila melanogaster* and called *yellow*. It was reported associated with larval pigmentation defects [24] and the yellowish pattern of defectives flies was only later explained by transposon-mediated mutagenesis and altered melanization enzyme activity [25]. Although all the members of the *yellow* gene family shared a conserved major royal jelly protein (MRJP) domain, not all *yellow* proteins can convert dopachrome into DHI, like *D. melanogaster yellow-f2* and *Aedes aegypti yellow* [26,27,28]. In the honey bee, *yellow* proteins are the main component (80%) of the royal jelly [29]. In addition to protective and antimicrobial functions, royal jelly is responsible of the larval development into queens [30], although the function of MRJPs in this process has yet to be elucidated [29,31]. *Yellow* genes have been reported as a gene family taxonomically restricted to insects, likely originating from horizontal gene transfer (HGT) from bacteria, and functionally diversified by species-specific duplications [32]. For instance, *yellow-like* transcripts found in the venom of a parasitoid wasp modulate host physiology and behavior [33], *yellow-like* salivary proteins of pathogen vectors, such as the sand flies *Phlebotomus tobbi* and *P. sergenti,* displayed leishmanicidal properties [34,35] and *yellow* proteins of the leaf beetle *Phaedon cochleariae* likely protect larvae from fungal infections [36]. Other *yellow-like* genes were identified in fungi and micro-eukaryotes, but their roles remains to be determined [37]. Apart from the aforementioned microscopic organisms and insects, *yellow-like* genes have been reported in few other species, namely in *Brachiostoma floridae* (Chordata), in the amoeba *Naegleria gruberi* and in the copepod *Lepeophtheirus salmonis* [32].

In bivalves, the enzymes involved in the biogenesis of melanin are often grouped into the general class of phenoloxidases, including three distinct enzyme types: tyrosinases (EC 1.14.18.1), catecholases (EC 1.10.3.1) and laccases (EC 1.10.3.2) [38], whereas less is known about the existence and functional role of mollusk dopachrome tautomerase genes.

The steady increase of sequenced genomes provides the opportunity for a comprehensive analysis of dopachrome tautomerase enzymes in metazoans. In this paper we investigated the taxonomic distribution as well as the phylogenetic relationships of DCT and DCE/*yellow* genes in genomic resources from a wider taxonomic distribution with a special focus on the lophotrochozoan phyla. By combining genome sequences with transcriptomic data, we could further demonstrate that these genes are functional and probably also underwent functional diversification outside of insects.

## 2. Materials and Methods

### 2.1. Retrieval and Processing of Sequence Datasets

All the predicted proteins of the Ensembl genome browser v95 (https://www.ensembl.org/index.html) and of the Ensembl Metazoa v42 (https://metazoa.ensembl.org/index.html) databases were downloaded from the corresponding ftp repositories (accessed in December 2018). To increase the representativeness of poorly supported lophotrochozoan clades, we retrieved the gene models of two gastropods (*Biomphalaria glabrata* and *Haliotis* spp.) [39], one brachiopod (*Lingula anatina*) [40], one nemertean (*Notospermus geniculatus*), one phoronid (*Phoronis australis*) [40] and 10 bivalve species (*Bathymodiolus platifrons*, *Modiolus philippinarum* [41], *Crassostrea gigas* [42], *C. virginica, Pinctada fucata* [43], *Ruditapes philippinarum* [44], S*accostrea glomerata* [45], *Mizuhopecten yessoensis* [46] *Limnoperna fortunei* [47] and *Scapharca broughtonii* [48]. We also analyzed the only genome available for copepods (i.e., *Tigriopus kingsejongensis*) [49] and one platyhelminth (*Schmidtea mediterranea*) [50]. Lancelets genomes were retrieved from JGI Genome Portal (*Branchiostoma floridae* [51]) and from the LanceletDB database (*B. belcheri*, http://genome.bucm.edu.cn/lancelet/index.php). RNA sequencing data for selected species were downloaded from the NCBI SRA archive and used for de-novo transcriptome reconstruction or for gene expression analysis (the list of species and additional details are reported in Appendix A). Raw reads were trimmed for quality using TrimGalore! (https://www.bioinformatics.babraham.ac.uk/projects/trim_galore/), allowing a minimal Phred quality of 30. For de-novo transcriptome reconstruction, the cleaned reads were assembled using CLC Genomic Workbench v11.0 (Qiagen, Hilden, Germany), applying a word size of 20; a bubble size of 50 and minimal contig length of 200 nt. The resulting contigs were subjected to open reading frame (ORF) prediction using the *transdecoder* tool implemented in the Trinity suite [52]. The details of the used sequence datasets are included in Appendix A.

### 2.2. Sequence Identification

The proteins of interest were identified from the protein datasets using archetypical domains, like the MRJP domain (Pfam ID: PF03022) to identify DCE/*yellow* proteins and the tyrosinase domain (Pfam ID: PF00264) for DCT proteins. These were coupled to *blast* searches using manually curated fungal, bacterial and insect *yellow* sequences retrieved from [32] or using the human DCT as query (UniProtKB ID P40126, TYRP2_HUMAN), with a *E-value* cut-off of 10^−10^. The conserved protein domains were identified using HMMer v3.1 [51], applying a *E-value* cut-off of 10^−5^. The presence of a signal peptide region was verified with Signalp v4.0 [53]. If the searches of the conserved domains were unproductive for a given genome, the whole genome was translated into amino acids in the six possible reading frames using the *transeq* tool (https://www.ebi.ac.uk/Tools/st/emboss_transeq/) and the presence of genomic regions corresponding to unannotated genes of interests was verified using HMMer, as described above.

### 2.3. Sequence Alignment and Phylogenetic Analysis

The cd-hit tool [54] was used to remove highly similar sequences (>90% of similarity). Protein sequences were aligned using the MUSCLE v3.8.31 tool with default parameters [55] and less informative alignment positions or sequences were identified and discharged using GUIDANCE [56], setting both position and sequence cut-offs to 0.6. ModelTest-NG v0.1.2 [54] was used to assess the best-fitting model of molecular evolution for the multiple sequence alignment (identified as the WAG+G+I+F model for DCT and as LG+G+I for DCE/*yellow*). Bayesian phylogenetic analysis was performed using MrBayes v3.2.6 [57]. Markov Chain Monte Carlo analysis were run with four chains for 10,000,000 generations (or until reaching the convergence cutoff of 0.01), with a sampling frequency of 1000 and a burn-in removal of 50% of the sampled trees. The convergence of parallel runs was estimated by reaching an average standard deviation of split frequency <0.05 and of a potential scale reduction factor equal to 1. Adequate posterior sampling was evaluated by reaching an effective sample size >200 for each of the estimated parameters using Tracer v1.6 [58]. The final majority-rule consensus tree was visualized and edited using FigTree v1.4.3 (http://tree.bio.ed.ac.uk/software/figtree/).

### 2.4. Re-Annotation of the *C. gigas* DCE/*Yellow* Gene Using RNA-Seq Data

To confirm the gene annotation of the *C. gigas* DCE/*yellow* gene (EKC25673) and to further investigate its expression levels, we sequenced two RNA libraries prepared from oysters from the Northern Adriatic Sea (Italy). In detail, oyster spat deployed in the lagoon of Goro or in a control tank in a local hatchery were sampled in June 2013. Soon after transportation to the laboratory, the oysters were dissected, the gills were placed in Trizol (ThermoFisher, Bremen, Germany) and stored at −80 °C. Total RNA was extracted following the Trizol producer’s recommendations. RNA quantity and quality were checked using a Nanodrop instrument (ThermoFisher) and an Agilent RNA6000 Nanochip (Agilent Technologies, Santa Clara, CA, USA), respectively. One microgram of high-quality RNA per sample was used to construct libraries for Illumina high-throughput sequencing, subsequently carried out with a 2 × 50 read layout in a Hi-Seq2000 instrument (Cornell Medical College, USA). Reads were trimmed as previously described and the clean reads were mapped on the *C. gigas* genome (ID: GCA_000297895.1) using the *large gap mapping tool* implemented in CLC (Qiagen, Hilden, Germany), applying 0.9 for both similarity and length parameters and the reads mapped around the oyster *yellow* locus (genome scaffold1670) were further taken into consideration. In parallel, the cleaned reads from each library were de-novo assembled, the *yellow* transcripts identified and compared with the genomic locus. Raw reads were deposited at the SRA archive under accession ID PRJNA484693.

### 2.5. Gene Expression Analysis of DCE/*Yellow* Genes

We used available RNA-seq datasets of selected non-insect metazoan species to compute the expression patterns of DCE/*yellow* genes in different tissues and conditions. These samples included 165 *C. gigas* datasets [42,59], 61 RNA-seq datasets from an experimental infection of *Mytilus edulis* with the parasitic copepod *Mytilicola intestinalis* [60], as well as other publicly available RNA-seq datasets, selected to investigate DCT/DCE expression patterns of species of interest (Appendix A). Clean RNA-seq reads were mapped on the corresponding genomes annotated with the predicted gene models or to the de-novo assembled contigs for the species without a sequenced genome, using the CLC read mapped, setting length and similarity fractions to 0.8 and 0.8, respectively, whereas mismatch/insertion/deletion penalties were set to 3/3/3. The number of unique mapped reads of each dataset were counted and used to calculate digital expression values as Transcripts Per Million (TPM), to ensure the comparability of different datasets [61].

### 2.6. Structural Analysis

The sequences of DCE/*yellow* proteins of *C. gigas*, *M. intestinalis*, *D. melanogaster*, *Octopus bimaculoides* and *B. belcheri* have been modeled by Swiss Model server (https://swissmodel.expasy.org) [62]. Other homology modeling servers (PSIPRED, PHYRE2 and I-TASSER) have been queried giving very similar results and have not been used for the subsequent analysis. The most robust and reliable results were obtained for the *D. melanogaster* DCE (sequence identity with PDB template 3Q6K: 28.41%) and *B. belcheri* (sequence identity with PDB template 3Q6K: 24.47%), while the other three models had a lower quality due to the low sequence identity levels (less than 20%) between queries and templates. The *D. melanogaster* DCE model, covering 84% of the submitted sequence (residues: 27–429), was built based on the target-template alignment using ProMod3 and fragments libraries to remodel insertions and deletions, and has been evaluated by GMQE function (global model quality estimation) and overall QMEAN (qualitative model energy analysis) scoring parameter, resulting in 0.60 and –3.28, respectively. The *B. belcheri* model, covering the 84% of the submitted sequence (residues: 33–388), has GMQE and QMEAN scores of 0.55 and −3.24, respectively. Models of *C. gigas M. intestinalis* and *O. bimaculoides* DCE/*yellow* proteins instead, covered 60% (residues: 122–725), 82% (residues: 29–400) and 82% (residues: 25–387) of the query sequence respectively, and resulted in GMQE <0.6 and of QMEAN <4. Geometrical parameters calculated for the resulting *D. melanogaster* DCE model (e.g., Interaction potential between Cβ: –3.13; interaction potential between all atoms: −3.19; solvation potential: –2.75; torsion angle potential: –2.24), which contribute to determine the global QMEAN, suffer the relatively low resolution of the best template exploited for the modeling (PDBID: 3Q6K).

## 3. Results

### 3.1. Taxonomic Distribution of Dopachrome Tautomerase Enzymes

Iterative searches performed on metazoan genomes and gene models allowed us to identify several putative dopachrome converting enzymes, namely DCT or DCE/*yellow* genes. To identify DCT proteins we used a combination of HMMer, *blastp* and signal peptide searches in order to select only the proper DCTs from all proteins sharing a *tyrosinase* domain (Figure 1). We found DCT genes in cephalopods, in one nematode (*Capitella teleta*), in platyhelminths, as well as in all the analyzed chordates, including uro- and proto-chordates (Table 1).

To map the taxonomic distribution of DCE proteins we used the whole *yellow* gene family, since it is not possible to discriminate between DCEs (the *yellow* proteins with a dopachrome tautomerase enzymatic activity) and other MRJP-domain containing proteins (called *yellow*, Figure 1). We started from 62 DCE/*yellow* sequences (58 are insect hits) reported by Ferguson and collaborators, who previously investigated the phylogenetic relationships of this gene family [32]. To this selection we could add DCE/*yellow* genes from some lophotrochozoan phyla, copepods and lancelets. Among lophotrochozoans, DCE/*yellow* genes are present in Mollusca, Brachiopoda and Rotifera, while they are apparently absent in Nematoda, Annelida, Nemertea and Phoronida (Table 1). Since the *phyla* of Nemertea and Phoronida are represented by two genomes only (*Notospermus geniculatus* and *Phoronis australis*, respectively), it is possible that DCE/*yellow* genes could be located on genome regions not included in the available assemblies or they could have been missed during gene annotation processes. To test both these hypotheses, we searched for the MRJP domain in Nemertea and Phoronida genomes translated into the six reading frames, as well as in transcriptomic data of these species (520 and 302 million reads, respectively). Since we failed to retrieve any MRJP-sequence with both methodologies, we deduced that, like for Annelida and Nematoda, the DCE/*yellow* gene is absent also in the lophotrochozoan phyla of Nemertea and Phoronida (Table 1).

We found evidence of the active expression of DCE/*yellow* genes in several lophotrochozoan species, for a total of 50 transcripts belonging to 33 species, including bivalves, cephalopods and gastropods. Moreover, we reconstructed the full-length DCE/*yellow* transcript of *O. bimaculoides* by de-novo assembly of one RNA-seq sample and we used it to correct the incomplete genome annotation. To further investigate the presence of DCE/*yellow* genes in non-insect metazoans, we systematically searched in *Branchiostoma* spp. and copepod genomic resources for the MRJP domain. We retrieved two DCE/*yellow* genes in both *B. floridae* and *B. belcheri* genomes, as well as for the recently genome-sequenced copepod *T. kingsejongensis*. We also identified 9 DCE/*yellow* transcripts searching in RNA-seq data of copepods.

In mollusk, we found one intronless DCE/*yellow* gene in most of the available genomes, while the other DCE/*yellow* genes are characterized by 2–5 introns. An intronless *yellow* gene was reported also in insects and it is referred as *yellow*-x [32,63]. In the *L. anatina*, *M. yessoensis* and *H. discus hannai* genomes, we found regions including two or more DCE/*yellow* genes organized as clusters, suggesting that recent duplication events have contributed to the increase of the number of DCE/*yellow* genes in these species. Taking into consideration the genes flanking the DCE/*yellow* loci of lophotrochozoans, we observed very few micro-syntenies, like the HES1 gene at the 3′ of *L. fortunei* and *M. philippinarum* genes or an overall high syntenic conservation for Ostreidae DCE/*yellow* genes (Appendix A).

### 3.2. Phylogenetic Analysis of *DCT* and *DCE*/yellow Proteins

We investigated the phylogenetic relationships of 83 aligned DCT proteins using a Bayesian phylogeny approach. The resulting tree clearly distinguished deuterostome from protostome DCTs. Within protostomes, lophotrochozoan hits clustered separately from platyhelminth DCTs. In the deuterostome cluster, cephalochordate and urochordate sequences formed a separate clade, which is located as an outgroup of the ‘conserved’ deuterostome DCTs (Figure 2). Additionally, we showed the presence of a clade including slightly different deuterostome DCTs.

The typical DCE/*yellow* protein encoded a single MRJP domain, with a N-terminal signal peptide region suggesting their extracellular localization (Figure 1). The presence of the single MRJP domain is common also in most of the bacterial and fungal proteins, although in few of them, the MRJP domain is associated with other domains (MFS_1, BAR_2, GFO_IDH_MocA, Med17, Iso-dh,). In some mollusk species we observed more variable domain organization. Ostreid bivalves, like *C. gigas*, *C. virginica*, *Saccostrea glomerata* and *P. fucata*, as well as few insects (*Bombus terrestris*, *Belgica antartica*, *Drosophila willistoni*, *Dendroconus ponderosa*, *D. plexuppus and L. cuprina*), possessed two MRJP domains in their DCE/*yellow* proteins. Two-domain DCE/*yellow* sequences were also retrieved from the bivalve transcriptomes of *Villosa lienosa* and *Margaritifera margaritifera* (family *Unionidae*). However, while the first group of bivalve species encoded only the two-domain DCE/*yellow* gene, the Unionidae species encoded both single- and two-domain genes. We found further support for this peculiar protein structure by back-mapping RNA-seq reads onto the genomic DCE/*yellow* locus of *C. gigas*. The even coverage along the whole gene suggests the complete transcription of two consecutive domains without an intron rather than an assembly artefact (Appendix A).

To study the phylogenetic relationships DCE/*yellow* proteins we considered a selection of MRJP protein domains including all the non-insect metazoan hits (116, including 62 transcriptomic-derived DCE/*yellow* sequences), a subset of the identified insect hits (obtained by reducing their redundancy to 0.7 using *cd-hit*, 271 sequences remained), 17 *yellow-like* protein sequences identified from insect’s salivary secretions, 7 MRJP-domain containing proteins of protozoans as well as 2 bacterial hits considered as possible outgroup.

The Bayesian phylogenetic tree (Figure 3), rooted on the node including protozoan and bacterial sequences, showed a clear separation between non-insect and most of the insect DCE/*yellow* sequences. The insect sequences (red branches in Figure 3) partially resembled the previous nomenclature of *yellow* protein types [32]. In fact, we obtained well separated clades for insect *yellow*-x, g, c and f, for a clade including most of the *yellow* proteins obtained from *Phlebotomus* spp. salivary secretions and for one clade including royal jelly proteins (MRJPs), with representatives of honey bee, *Atta cephalotes*, *B. terrestris*, *Nasonia vitripennis* and *Solenopsis invicta*. As previously reported [32], y*ellow*-x type appeared as the ancestral insect’s *yellow,* with the other DCE/*yellow* types probably originating from x-type ancestors by taxon-specific functional diversification and expansions.

Similar to *yellow*-x, copepod and cephalopod hits (highlighted in violet and light blue, respectively) appeared ancestral, although their phylogenetic position relative to the *yellow*-x group and other mollusk proteins remained unresolved. Lancelet DCE/*yellows* (highlighted in green) clustered as outgroup of metazoans, together with a *Reticulomyxa filosa* hit (Foraminifera), possibly indicating an independent horizontal gene transfer (HGT) event.

DCE/*yellow* are widespread within mollusks and they clustered in several clades according to their taxonomic group. Gastropod DCE/*yellow* sequences formed two distinct clades, while bivalve hits clustered in three clades. When species had multiple copies of DCE/*yellow* (e.g., *Mizuhopecten yessoensis, H. discus*), these tended to cluster together, possibly indicating recent duplication events. The first bivalve sub-cluster (highlighted in yellow) comprises most of the sequences (33 out of 57) and it included one sequence for almost all the analyzed species. The second and third sub-clusters included sequences of species showing DCE/*yellow* gene expansions, like *M. yessoensis*, Mytilidae and Unionidae species. The two-MRJP domains of double-domain DCE/*yellow* genes of each species clustered into the same clade, for both bivalve and insect proteins, possibly indicating incomplete duplication events.

### 3.3. Structure of DCE/*Yellow* Proteins

It is difficult to assess the functional relevance of genes from sequence data alone. One possibility to gain the functional importance is to analyze the structural conservation and transcription patterns of the genes in question.

To investigate differences and similarities between the structure of DCE/*yellow* proteins, we build up structural models of *C. gigas*, *M. intestinalis, D. melanogaster*, *O. bimaculoides* and *B. belcheri* proteins, based on to two similar structures identified in the Protein Data Bank (PDB) database: the salivary protein LJM11 from *L. longipalpis* (PDB ID: 3Q6K) and the MRJP1 from *Apis mellifera*. (PDB ID: 5YYL). Due to the low sequence identity (around 20%), only the structural models of *D. melanogaster* and *B. belcheri* (respectively 28% and 24% of identity with *L. longipalpis* salivary protein) produced reliable scores (see Materials and Methods, Section 2.6). Despite the low structure scores, all the five modeled proteins shared the same conserved 6-bladed β-propeller fold (Figure 4A), irrespective of the used software (SWISS-MODEL, PSIPRED, PHYRE2 and I-TASSER). As for MRJP1 and LJM11, whose structures have been already solved [64,65], the structures showed that the position of the core six-bladed β-propeller should be similar, while the peripheral bucket is somewhat more variable in terms of conserved residues and of their spatial arrangements. Regarding the peripheral part at the entrance of the pore, *D. melanogaster* and *M. intestinalis yellow* proteins seemed to share typical features of LJM11 (Figure 4B). In particular, in both proteins aromatic residues (sometimes mutated in residues with similar characteristics e.g., Phe-Tyr) that are involved in the binding of the indole portion of serotonin in LJM11 were conserved [64]. These were: Tyr-90, Phe-178, Phe-223, Phe-325 and Phe-344, that correspond to Tyr-143, Trp-265, Phe-270, Tyr-378 and Phe-400 in *D. melanogaster yellow* and Phe-127, Phe-151, Phe-193, Tyr-245 and Trp-366 in *M. intestinalis* DCE/*yellow* protein (Table 2). Moreover, two residues were present in LJM11 that were located inside the pore and divided it into two pockets (Gln-91 and Asp-328) and are probably involved in the interaction with the phenolic hydroxyl group of serotonin (Gln-91 only) [64]. Interestingly, these two residues are conserved in *M. intestinalis yellow* protein (Gln-129 and Asp-368), while only the aspartate residue is conserved in *D. melanogaster* DCE (Asp-381), since the glutamine residue is substituted by an arginine (Arg-144). DCE/y*ellow* proteins of *O. bimaculoides* and *B. belcheri* instead, share only few of these aromatic residues (two for *O. bimaculoides* and just one for *B. belcheri*, see Table 2). Both these two proteins share an aromatic residue that corresponds to the Phe-325 of *L. longipalpis* LJM11 (respectively His-333 in *O. bimaculoides* and Trp-344 in *B. belcheri*), that is also present in a putative hydrolase protein of the bacterium *Anabaena variabilis* (Trp-291). Moreover, the *A. varaiabilis* protein shares the peculiar 6-bladed β-propeller fold (X-ray structure, PDBID: 2QE8) and moderate sequence identity with *B. belcheri* and *O. bimaculoides* DCE/*yellow* proteins (14% with *O. bimaculoides* and 26% with *B. belcheri*).

For the *C. gigas* DCE/*yellow* protein, the structures of both MRJP domains were poorly predictable from a structural point of view. In fact, besides the 6-bladed β-propeller fold and a few conserved residues, it was very difficult to find similarities to MRJP1 or LJM11. Even the C-terminal domain, between the two MRJP domains, representing the one with the highest identity with *A. mellifera* MRJP1, only had a similarity of 24.58%, while the N-terminal domain only showed 23.10% of similarity.

### 3.4. Expression of DCE/*Yellow* Genes in Non-Insect Species

Within the transcriptomes of species containing DCE/*yellow* genes, we found substantial variation in their expression values, ranging from zero to thousands of TPMs (Figure 5). We reported that 33% of the DCE/*yellow* expression values are under 10 TPMs, considered as a cut-off for low-expressed genes, whereas 5% of them exceeded 100 TPMs. Among these we found four *H. rufescens* (gastropod) expression values related to one of its three DCE/*yellow* genes in kidney and liver tissues, four *O. bimaculoides* values (cephalopod, skin, suckers and visceral mass tissues) as well as 18 out of the 20 *M. intestinalis* values (Crustacea, Copepoda). Other copepod species, like the free-swimming copepod *T. kingsejongensis* and the ectoparasitic salmon louse (*L. salmonis*), showed lower DCE/*yellow* expression values when compared with *M. intestinalis*. Bivalve DCE/*yellow* genes appeared poorly expressed, with the highest expression levels in gill, mantle and gonad tissues. Finally, the two *B. belcheri* DCE/*yellow* genes are expressed at intermediate levels during developmental stages (Figure 5).

These results suggested that DCE/*yellow* of *M. intestinalis* could retain a role during *Mytilus-Mytilicola* parasitic interactions and makes this combination of hosts and parasites both possessing DCE/*yellow* genes an interesting case to study functional gene diversification. However, while mussel DCE/*yellow* showed low transcription in all the samples of mussel gut tissue, in all the *Mytilicola* samples DCE/*yellow* was among the top100 expressed genes.

For *C. gigas* we obtained detailed information on the timing and tissue location of DCE/*yellow* expression, including the two gill RNA-seq datasets produced for this study (Appendix A). We found that oyster DCE/*yellow* is preferentially expressed in gills, whereas it is poorly expressed in the other tested tissues (Figure 6A). During ontogeny DCE/*yellow* is not expressed in the first developmental stages, but its expression increases constantly after the D-shaped larval stage (Figure 6B) and infection with *Ostreid herpesvirus-1* (OsHV-1) was downregulating DCE/*yellow* up to 3.2-fold at 24 h post infection (Figure 6C). The tissue specificity and constitutive expression in gills after reaching a certain developmental stage strongly suggest that this gene has functional relevance for oysters.

Finally, since both octopus and lancelet encoded also DCT genes, we included their expression profiles in Figure 5. We reported that, among the tested samples, *O. bimaculoides* DCT is expressed quite exclusively in the retina, while *B. belcheri* DCTs showed intermediate expression levels along developmental stages.

## 4. Discussion

Melanin is widespread in the animal kingdom, where it is involved in several important biological functions including pigmentation and immune-defense [6,8,66]. Here, we now traced the distribution of the DCT and DCE/*yellow* gene families through metazoan evolution. DCTs possess the capacity to convert dopachrome into DHICA [17,18], while some DCE/*yellow* proteins can convert dopachrome into DHI. Both DHICA and DHI, combined in different ratios, are needed to build eumelanin [6,15], and their enzymatic conversion is considered to be the bottleneck of melanin production [19].

Both gene families appeared as highly dynamic, whose distribution is characterized by extensive gene losses and, possibly, horizontal gene transfer (HGT) events. We showed that DCTs are mainly found in deuterostomes and only in a few non-deuterostome organisms, including cephalopods, platyhelminths, and *C. teleta* as only representative of nematodes (Table 1). As eumelanin represents the dark pigment in cephalopod ink [6] and in the black eye spots of platyhelminth species [67], DCTs seem to play a crucial role for dark coloration in a wide range of taxa. However, eumelanin is also widespread in insects and other invertebrates [17], most of which do not encode a DCT gene. Indeed, melanization is a normal process during shell formation in mollusks [68].

This suggests that the enzymatic function of DCT (dopachrome tautomerase) is performed by a different protein, with DCE/*yellow* being a likely candidate [17]. The DCE/*yellow* family was previously thought to be exclusively found in insects, bacteria and fungi [29,32]. Our results now suggest that along with the early-diverging metazoan phyla of Placozoa, Porifera, Cnidaria, and Ctenophora, only the lophotrochozoans Annelida, Nemertea and Phoronids do not possess these genes. In contrast, we found that almost all other analyzed lophotrochozoans have one or more DCE/*yellow* genes, while cephalopods and lancelets even possess both DCT and DCE/*yellow* genes (Table 1).

This patchy distribution suggests that DCT and especially DCE/*yellow* genes were already present in the protostome and deuterostome ancestor and both genes could have experienced extensive gene loss events. DCE/*yellow* was then lost in deuterostomes, in Malacostraca among arthropods as well as in the lophotrochozoan phyla mentioned above, whereas DCT has been lost in most protostomes, with the exception of cephalopods. The presence of a DCT gene in a single nematode species (*C. teleta*), as well as the presence of DCE/*yellow* genes in lancelets can possibly be explained by HGT events. Prokaryote-to-Eukaryotic HGT is considered to be an important element driving eukaryote genome evolution [69,70], whereas Eukaryote-to-Eukaryote HGTs were only seldom reported (e.g., in the foraminifera *R. filosa* [71]). Curiously, in the phylogenetic tree DCE/*yellow* proteins of lancelets were closely related to the DCE/*yellow* of *R. filosa*, possibly indicating an HGT event (Figure 3).

This extensive gene loss scenario might be feasible due to the functional redundancy of DCT and DCE with either gene being able to compensate the loss of the other in the production of melanin. Similarly, both *D-dopachrome tautomerase* (D-DT) and its paralogue *Macrophage migration inhibitory factor* (MIF) also possess a similar enzymatic capacity, although on the non-physiological substrate D-dopachrome [72,73]. When honey bee *Major Royal Jelly Protein 1* is artificially expressed in mice the production of D-DT is decreased [74], indicating a possible potential compensative role between these two proteins. We recently investigated the taxonomic distribution of D-DT genes demonstrating a patchy distribution among protostomes, with gene losses in insect and crustacea as well as in the bivalve *C. gigas* ([75] unpublished data until accepted). Although the hypothesis that DCT, DCE/*yellow* and D-DT gene losses could be functionally compensated, this hypothesis requires experimental proof and, on the basis of available knowledge, it remains a speculation.

The presence of a gene or conserved domain does however not demonstrate that it is functional. One line of evidence that might support functionality comes from the expression of non-insect DCE/*yellow* genes. Although we retrieved abundant RNA-seq datasets for some mollusk species, DCE/*yellow* expression levels show often limited to very limited expression values (Figure 5). Among bivalves, we showed a preferential of *C. gigas* DCE/*yellow* expression in gills and mantle tightly timed with ontogeny (Figure 6). DCE/*yellow* expression was also substantially down regulated in the oyster gills after OsHV-1 infection suggesting a virus-mediated modulation of the host’s DCE/*yellow* pathway. This would support that one of the pleiotropic functions exerted by insect DCE/*yellow* proteins is the modulation of host-parasite interactions [34,64]. In this context, the *Mytilus*-*Mytilicola* host-parasite system represents the only situation in which both host and parasite encode a DCE/*yellow* gene. While the mussel host did hardly express its DCE/*yellow* gene in the gut, the gene of the parasitic copepod was among the most abundant transcripts (Figure 5). Such high expression level possibly represent a response to the challenges posed by the gut environment and to the release of ROS by the mussel host [60], an hypothesis consistent with the known ROS scavenger function of melanin [66]. In the free-swimming copepod *T. kingsejongensis* as well as in all developmental stages of the ectoparasitic salmon louse *L. salmonis*, on the other hand, we found only low expression levels of their DCE/*yellow* genes further suggesting species-specific functional diversification and a prominent role of DCE/*yellow* proteins limited to the *Mytilus*-*Mytilicola* host-parasite interaction. Arguably, also the salivary MIF of ticks and aphids are involved in the modulation of host-pathogen interactions [76], intriguingly mirroring the role of insect’ salivary yellows [34,35,64]. Therefore, functional validations are definitively needed.

The active expression of *B. belcheri* DCE/*yellow* genes also suggests that it serves a functional role although this gene might have originated from HGT. Finally, we found that in *O. bimaculoides* as one of the few species possessing both DCT and DCE/*yellow* genes, DCT is exclusively expressed in the retina and DCE/*yellow* in the skin tissues. In humans, both these tissues showed high expression levels of DCT [77,78] and this intriguing similarity might suggest that both octopus DCT and DCE/*yellow* genes serve functional roles in different tissues. Structural modelling of both octopus and lancelet DCE/*yellow* proteins revealed an overall structural similarity with insect’s *yellow*, although the conservation of key residues located inside the putative enzymatic pocket is low. This latter result, in particular for the octopus DCE/*yellow* protein is striking, since the high expression in skin tissues strongly suggests its involvement in the melanogenic pathway. Interestingly, both these two proteins showed a conserved aromatic residue, that is also present in a putative hydrolase protein of the bacterium *A. variabilis*, further support the connection between bacterial and metazoan DCE/*yellow* genes.

Ferguson and collaborators [32] proposed that the metazoan *yellow* gene family originated from an HGT from bacteria. However, previous phylogenetic analysis failed to identify a clear link between bacteria and insect *yellow* genes, since the co-clustering of bacterial and of *yellow*-x hits, the most ancestral clade of insect’s *yellows*, greatly depended by the parameters adopted for the analysis [32,63]. We are aware that, also in our analysis, the high heterogeneity of DCE/*yellow* genes, coupled with the small number of informative sites, somewhat limited the resolution of the phylogenetic tree. However, by increasing the number of the originally sampled species, we showed a clear distinction between the DCE/*yellow* sequences of the different animal phyla (Figure 3). Among the cluster of each phylum, the DCE/*yellow* sequences only partially followed the phylogenetic relationships among species, indicating the presence of different functional subtypes shared between species of the same phyla. This aspect is particularly evident for insect genes, where this gene family underwent a rapid evolution, resulting in high sequence diversification and remarkably different functional adaptations between and within species [26,32,63]. Likewise, we reported that also mollusk DCE/*yellow* genes experienced considerable diversification in some species, like in *M. yessoensis* (11 genes), in Mytiloida species (3–5 genes) and in gastropod species (*Haliotis* spp. and *B. glabrata*). Although we could trace one intronless DCE/*yellow* in several lophotrochozoan species, we observed a considerable syntenic conservation only for species of the same family (e.g., Ostreoida, Appendix A). Low conservation of the flanking genes was also reported for insect DCE/*yellow*, including the low conservation of their cys-regulatory landscape [79], further supporting the dynamic nature of evolution in this gene family.

The peculiar DCE/*yellow* protein structure of ostreid bivalves nicely illustrates these evolutionary dynamics. In oysters DCE/*yellow* is composed of two MRJP domains (Appendix A), and the high similarity of these two MRJP domains, along with the lower conservation of key residues as well as of the overall structural features possibly indicate a recent incomplete duplication followed by functional liberation and random drift through mutational and functional space. Moreover, the presence of marked structural differences of the *C. gigas* protein, taken as example for bivalve DCE/*yellow* proteins, can suggest that these genes have evolved different functions starting from the same structural core (Figure 4).

This structural gene diversification might suggest that melanin production is not the only function served by DCE/*yellow* genes. Indeed, while all DCTs can convert dopachrome into DHICA [17,18], only a subset of the genes described as DCE/*yellow* can convert dopachrome into DHI. Other DCE/*yellow* genes are involved in cuticle and eggshell formation and hardening during insect development, protection from dehydration and, as salivary proteins, they confer protective immunity against *Leishmania major* as well as they can bind several ligands, including serotonin [26,29,32,63,64]. The structural analyses of the *M. intestinalis* DCE/*yellow* protein revealed shared common features with *D. melanogaster* DCE, and such features were also present in the salivary *yellow* proteins of *Phlebotomus* but not in the *A. mellifera* MRJP structural model (Figure 4). Given these similarities, we could be tempted to speculate that these two proteins may retain an activity similar to that of salivary proteins, but in-vitro experiments with recombinant proteins are needed to demonstrate these functions.

## 5. Conclusions

In the present work we provided a wide-ranging taxonomic distribution of dopachrome tautomerase genes in metazoans. While we confirmed that DCT is mostly distributed among deuterostomes, we reported a wide distribution the DCE/*yellow* gene family in protostomes, reverting the previous data reporting that *yellow* is a gene family taxonomically restricted to insects. The most probable scenario for this taxonomic distribution is that an ancestral DCE/*yellow* gene was present before the radiation of proto/deuterostomes and subsequent extensive gene loss events have occurred among several animal phyla. Taken together these observations suggested that the dopachrome conversion enzymatic activity can be performed complementary by different unrelated enzymes and this complementarity might have released selective pressures and enabled the diversification (insects, mollusks) or loss (e.g., crustaceans other than copepods, Nematoda, Annelida) of DCE/*yellow* genes. The analysis of transcriptomic samples suggested phyla-specific functional adaptations, although, given the importance of melanisation in the immune response, it seems conceivable that these genes also may have maintained the dopachrome converting enzymatic activity. Future studies, leveraging on recombinant proteins and/or gene silencing could help unraveling the detailed functions of DCE/*yellow* in lophotrochozoans.

## Figures and Tables

**Figure 1 genes-10-00495-f001:**
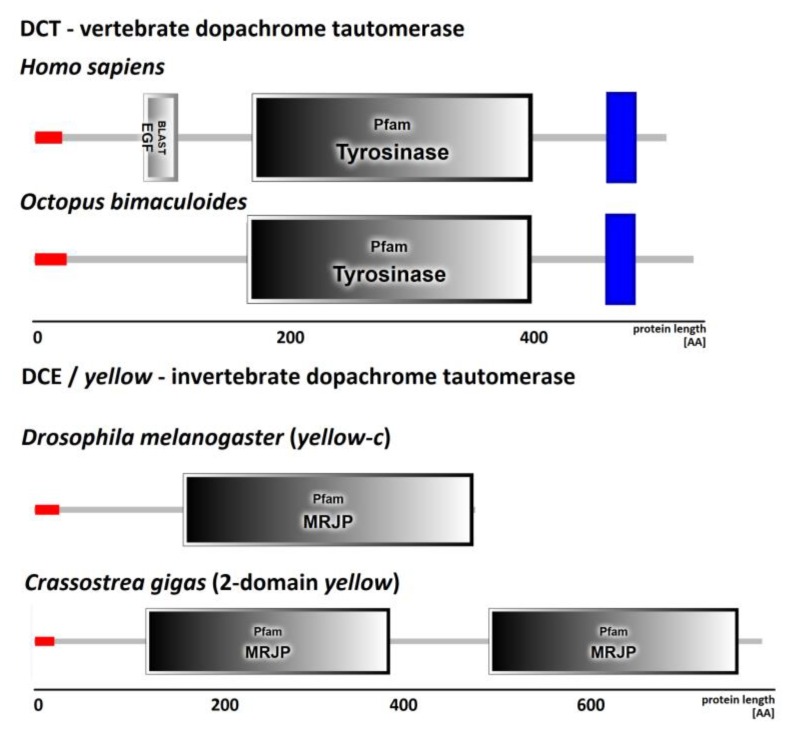
Dopachrome tautomerase (DCT) and Dopachrome converting enzyme (DCE) protein structures showing the conserved domains and motifs found in *Homo sapiens* and *Octopus bimaculoides* DCTs as well as in *Drosophila melanogaster* and *Crassostrea gigas* DCE/*yellow*. Red boxes depict signal peptide regions, blue boxes transmembrane motifs and gray rectangles the conserved PFAM domains (EGF, PF00008; Tyrosinase, PF00264; MRJP, PF03022).

**Figure 2 genes-10-00495-f002:**
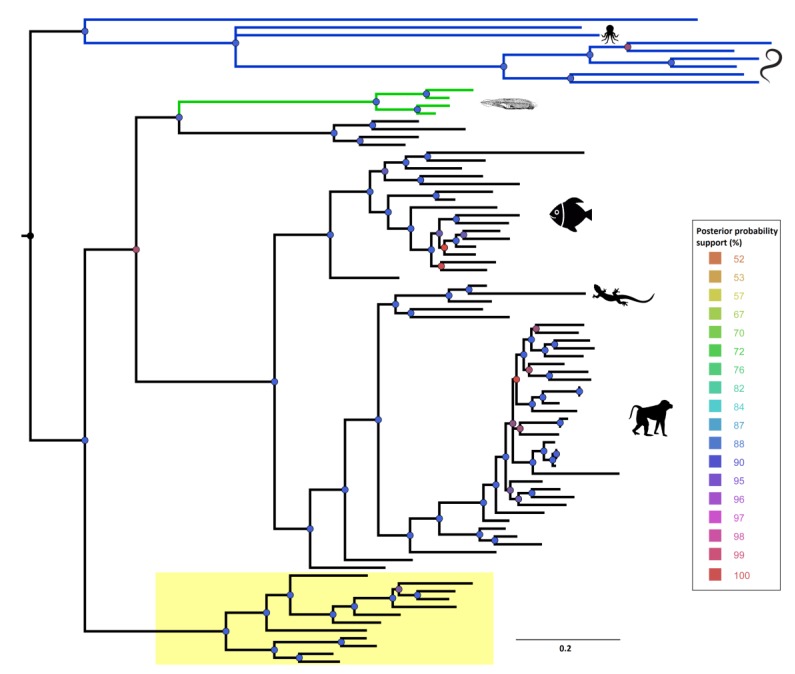
Bayesian phylogenetic tree of metazoan DCT proteins. Blue lines referred to protostome sequences and green lines to lancelet sequences. The box highlighted in yellow included a different type of vertebrate DCT, present in few species. The posterior probability support was reported for each node in a color-scale, ranging from 50 to 100% of support. Icons of representative species were drawn for the main clades. The phylogenetic tree and the multiple alignment are included in Appendix A.

**Figure 3 genes-10-00495-f003:**
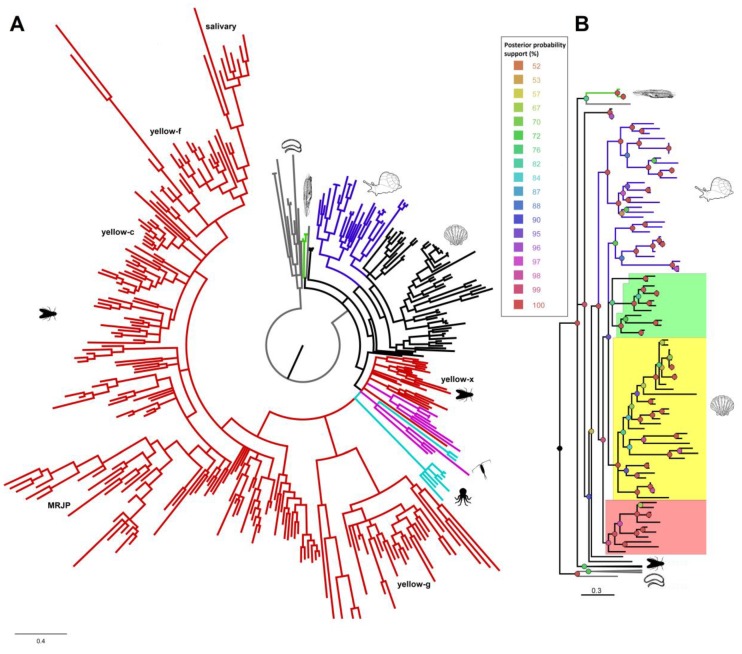
Bayesian phylogenetic tree obtained from the alignment of metazoan DCE/*yellows*. **A.** Insect sequences are shown in red, copepod sequences in violet, cephalopod sequences in light-blue, black lines denote bivalve hits, blue lines refer to gastropod hits and green lines refer to lancelet proteins. Protozoan and bacterial *yellow-like* proteins used as outgroup (shown in grey). The different insect groups are marked according to their previous nomenclature (*yellow*-x, g, c, f, MRJP and salivary proteins). **B**. The cladogram zooms into the non-insect DCE/*yellow* hits, using the same color-codes. Posterior probability support is reported for each node in a color-scale, ranging from 50 to 100% of support. Icons of representative species show taxonomic groups. The phylogenetic tree and the multiple alignment are included in Appendix A.

**Figure 4 genes-10-00495-f004:**
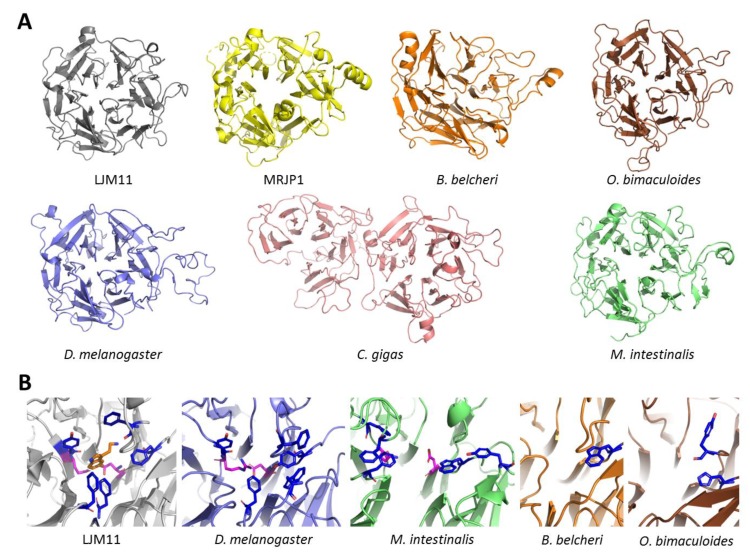
Comparison between the structure of LJM11 (PDB ID: 3q6k), MRJP1 (PDB ID: 5yyl) and the structure models of DCE/*yellow* proteins *of B. belcheri*, *O. bimaculoides*, *D. melanogaster*, *C. gigas* and *M. intestinalis*. Panel A: general view of the 6-bladed β-propeller fold of the templates and the models. Panel B: detail of the pore entrance of LJM11 compared to the DCE/*yellow* proteins *of D. melanogaster, M. intestinalis*, and *B. belcheri*. The serotonin molecule is colored in orange, the aromatic residues and the conserved Gln/Arg-Asp residues that divide the pores in two pockets are shown in blue and magenta, respectively.

**Figure 5 genes-10-00495-f005:**
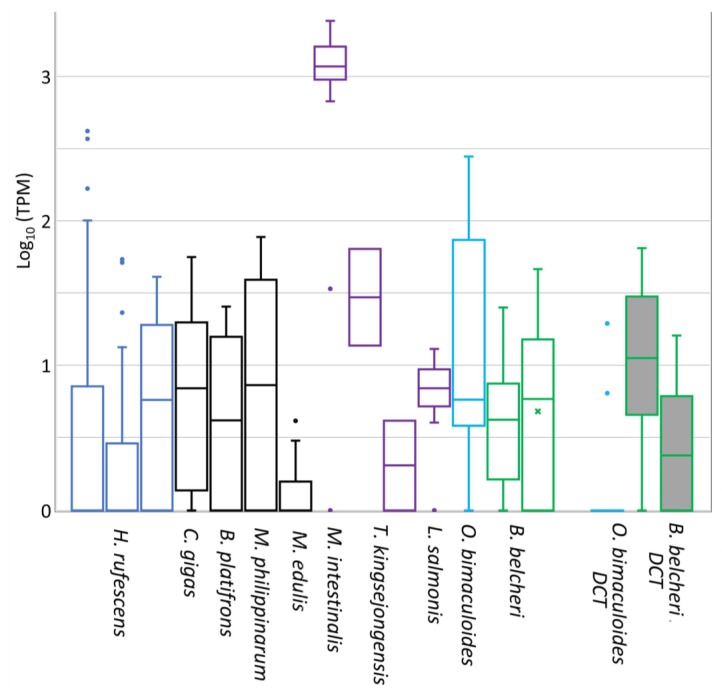
Box plot of the expression values of DCE/*yellow* genes in non-insect metazoans. Expression values are reported as log transformed Transcript Per Million (TPM) for the gastropod *H. rufescens* (3 genes, 40 samples); for the bivalves *C. gigas* (165 samples), *B. platifrons* (6 samples), *M. philippinarum* (5 samples), *M edulis* (41 samples); the copepods *M. intestinalis* (20 samples), *T. kingsejongensis* (2 genes, 2 samples), *L. salmonis* (50 samples); the cephalopod *O. bimaculoides* (19 samples) and the lancelet *B. belcheri* (2 genes, 18 samples). Additionally, the expression of DCT genes was reported for the two species encoding this gene (*O. bimaculoides* and *B. belcheri*, boxes filled in grey).

**Figure 6 genes-10-00495-f006:**
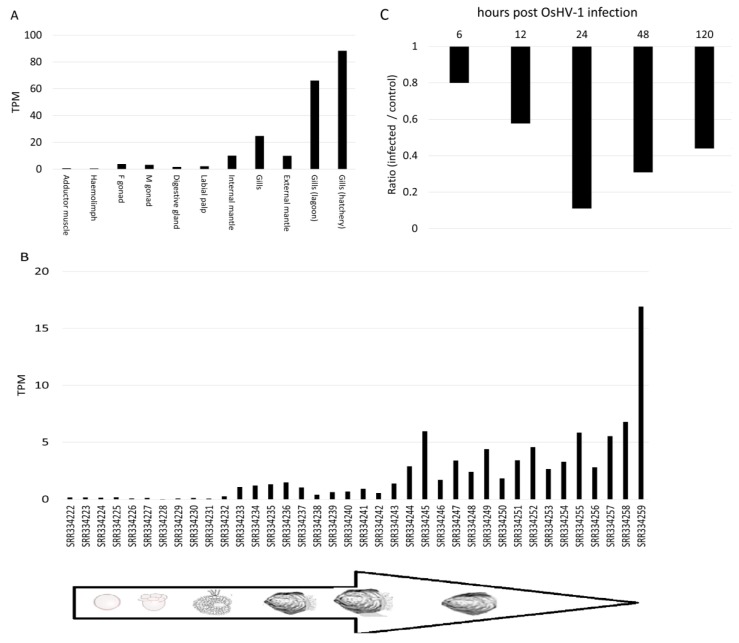
Expression levels of the *C. gigas* DCE*/yellow*. The expression values are reported as TPM. (**A**) Different oyster tissues [42], including the two gill RNA-seq samples described in this work; (**B**) during ontogeny [42]; (**C**) along a 0–120 hours’ time-course OsHv-1 infection experiment [59], in this case reported as fold change of treated versus control samples.

**Table 1 genes-10-00495-t001:** Taxonomic distribution of DCT and DCE/*yellow* genes in Metazoa. Group, phylum, class, number of DCT and DCE/*yellow* genes per representative species are reported.

Group	Phylum	Class	DCT	DCE (Gene)	DCE (Transcript)
Lophotrochozoa	Mollusca	Bivalvia	no	X	X
		Gastropoda	no	X	X
		Cephalopoda	X	X	X
	Annelida		no	no	nt
	Nemertea	Pilidiophora	no	no	no
	Brachiopoda	Lingulata	no	no	nt
		Phoronida	no	no	no
	Rotifera	Bdelloidea	no	no	nt
Ecdysozoa	Nematoda	\	X *	no	nt
	Arthropoda	Malacostraca	no	no	nt
		Hexanauplia	no	X	X
		Collembola	no	X	nt
		Insecta	no	X	nt
	Platyhelminthes		X	no	nt
Deuterostomia	Chordata	Ascidiacea	X	no	nt
		Branchiostomidae	X	X	nt
		Craniata	X	no	nt

Y: dataset analyzed with positive hits; N: dataset analyzed without positive hit; nt: not tested. * only in one of the tested species (*C. teleta*).

**Table 2 genes-10-00495-t002:** Residue correspondence between *L. longipalpis* LJM11 and the DCE/*yellow* proteins of *D. melanogaster*, *M. intestinalis*, *O. bimaculoides*, *B. belcheri and A. variabilis*.

*L. longipalpis*	*D. melanogaster*	*M. intestinalis*	*O. bimaculoides*	*B. belcheri*	*A. variabilis*
Tyr-90	Tyr-143	Phe-127	---	---	---
Phe-178	Trp-265	Phe-151	---	---	---
Phe-223	Phe-270	Phe-193	---	---	---
Phe-325	Tyr-378	Trp-366	His-333	Trp-344	Trp-291
Phe-344	Phe-400	Tyr-245	Tyr-352	---	---
Gln-91	---	Gln-129	---	---	---
Asp-328	Asp-381	Asp-368	---	---	---

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
