# Peer review of "An Evolutionary Perspective of Dopachrome Tautomerase Enzymes in Metazoans"

_genes, 2019, doi:10.3390/genes10070495_

Round 1

Reviewer 1 Report

Dopachrome tautomerase (DCT) and dopachrome converting enzyme (DCE) play important roles in the production of melanin, which, in turn, plays an important role in innate immune defenses in metazoans. The authors have carried out a comprehensive taxonomic analysis of these enzyme in metazoans and find that they are much more widespread than previously reported (and not limited to insects, bacteria, and fungi).  The analysis is carefully and rigorously done.  Publication is recommended after the authors consider the following points.   

I’m struck by the parallels between DCT and MIF in the combinations of hosts and parasites (lines 388-390, lines 451-452, lines 544-545). There’s a recent paper that does a similar analysis of MIF and makes a similar conclusion (Sparkes, A., De Baetselier, P., Roelants, K., De Trez, C., et al., 2017, “The non-mammalian MIF superfamily”, Immunobiology 222, 473-482.) Maybe the author could discuss their findings in the context of the MIF findings?

Some minor questions and typographical issue -

Lines 334-336 – Does DCT have an N-terminal proline (like MIF)?  If so, is it conserved?

Line 52 – I’m not sure rarely investigated is the correct terminology.

Line 64 – In addition to

Line 74 – aforementioned

Line 120 – amino acids are two words

Line 122 – Change not annotated to unannotated

Line 222 – evidence

Line 251 – as an outgroup

Line 456 – we recently investigated

Author Response

Dear Editor Marcos Arranz,

We are grateful to you and the two anonymous reviewers for the positive and useful comments on our study concerning the phylogenetic distribution of “yellow” genes. We carefully considered all the raised points and, accordingly, we modified some sentences in the manuscript. Below, the point-by-point answers to reviewer’s questions.

We hope that our revisions will be deemed satisfactory and that our paper can be accepted for publication.

Best Regards,

On behalf of all co-authors,

Umberto Rosani

Response to Reviewer 1.

Dopachrome tautomerase (DCT) and dopachrome converting enzyme (DCE) play important roles in the production of melanin, which, in turn, plays an important role in innate immune defenses in metazoans. The authors have carried out a comprehensive taxonomic analysis of these enzyme in metazoans and find that they are much more widespread than previously reported (and not limited to insects, bacteria, and fungi).  The analysis is carefully and rigorously done.  Publication is recommended after the authors consider the following points.   

I’m struck by the parallels between DCT and MIF in the combinations of hosts and parasites (lines 388-390, lines 451-452, lines 544-545). There’s a recent paper that does a similar analysis of MIF and makes a similar conclusion (Sparkes, A., De Baetselier, P., Roelants, K., De Trez, C., et al., 2017, “The non-mammalian MIF superfamily”, Immunobiology 222, 473-482.) Maybe the author could discuss their findings in the context of the MIF findings?

Thanks for your positive comments. According with the conclusions of Sparkes and colleagues on MIF/D-DT genes, we discussed possible similarities with DCE/yellow in modulating host-pathogen interactions (see lines 483-485). However, as correctly pointed out by Reviewer2, we should clearly state that these are speculations, since functional validations are needed to support this link (see lines 461-464). DCT do not possess a conserved N-proline, as the one present in MIF and D-DT, where it play a fundamental role for enzymatic activity.   

Thanks, we have fixed all the minor issues and typos listed below.

Some minor questions and typographical issue -

Lines 334-336 – Does DCT have an N-terminal proline (like MIF)?  If so, is it conserved?

Line 52 – I’m not sure rarely investigated is the correct terminology.

Line 64 – In addition to

Line 74 – aforementioned

Line 120 – amino acids are two words

Line 122 – Change not annotated to unannotated

Line 222 – evidence

Line 251 – as an outgroup

Line 456 – we recently investigated

Reviewer 2 Report

The study describes the curation and analysis of a large collection of sequences identified as dopachrome tautomerase (DCT) and dopachrome converting enzyme (DCE) from publicly available sequence data from metazoans. A major finding is that DCEs, previously reported to be restricted to insects, bacteria and fungi, are shown to be present and functional in most of the lophotrochozoan phyla and in copepods. Authors then performed a Bayesian phylogeny of both DCT and DCE, compared the expected 3D structure of a set of DCE proteins, and inferred the expression of DCE genes in non-insect species from the analysis of RNA-seq datasets.  

This study should be of particular interest to those studying DCT and DCE. A general concern is that the functional importance of these enzymes is unclear and their link with immune response or host-parasite interaction appears over-interpreted.

For example:

- Introduction section: L35-37: Authors say “The production of a capsule surrounding the invader is followed by its melanization to enclose the invader into an environment with high concentration of ROS [3].”  The mounting of a melanized capsule is a well know defense response of arthropods but for most of other invertebrate phyla the encapsulation is not followed by melanization. The link between melanization and immunity is therefore not that clear (or not existing) for most invertebrates and this should be clarified.

- L388-390: Authors observe a higher expression level of DCE in a parasitic copepod as compared with free living copepods and state that “These results suggested that DCE/yellow of M. intestinalis retains a primary role during parasitic interaction and makes combinations of hosts and parasites both possessing DCE/yellow genes interesting cases to study functional gene diversification". Such a general conclusion on the potential importance of DCE would require to observe a higher expression level in several parasitic species.

- In the discussion authors argue that “gene loss might be functionally compensated among DCT, DCE/yellow and D-DT in multiple ways”. This is indeed an interesting idea. However, a potential functional redundancy between DCT or DCE and D-DT remains highly speculative and this should be clarified in the text. Although D-DT (and MIF too, this should be corrected L 453) show a D-dopachrome tautomerase activity, the natural substrate remains unknown and their involvement in melanization has not been functionally demonstrated despite the impressive number of functional studies on these proteins.

Minor points:

- the use of “orthologues” and “paralogues” should be avoided throughout the manuscript as, in most cases, this terminology is not supported by data (e.g. L198, L411, L429)

- L102-104: Clarify the fact that the list of selected species is provided in the supplementary data.

- L 150: What are clean reads here? Please, clarify.

-L 156: It may be my mistake but the SRA archive under accession ID PRJNA484693 was not accessible to me. 

- L441: “combined with multiple, but few HGTs” is somehow contradictory 

- HGT: Authors mention the existence of HGT in several instances, but it remains unclear to me what are the evidences of HGT. For example, the positions of the genes -supposed to originate from HGT- in the phylogenetic do not support this statement. Please clarify.

Author Response

Dear Editor Marcos Arranz,

We are grateful to you and the two anonymous reviewers for the positive and useful comments on our study concerning the phylogenetic distribution of “yellow” genes. We carefully considered all the raised points and, accordingly, we modified some sentences in the manuscript. Below, the point-by-point answers to reviewer’s questions.

We hope that our revisions will be deemed satisfactory and that our paper can be accepted for publication.

Best Regards,

On behalf of all co-authors,

Umberto Rosani

Response to Reviewer 2.

The study describes the curation and analysis of a large collection of sequences identified as dopachrome tautomerase (DCT) and dopachrome converting enzyme (DCE) from publicly available sequence data from metazoans. A major finding is that DCEs, previously reported to be restricted to insects, bacteria and fungi, are shown to be present and functional in most of the lophotrochozoan phyla and in copepods. Authors then performed a Bayesian phylogeny of both DCT and DCE, compared the expected 3D structure of a set of DCE proteins, and inferred the expression of DCE genes in non-insect species from the analysis of RNA-seq datasets.

This study should be of particular interest to those studying DCT and DCE. A general concern is that the functional importance of these enzymes is unclear and their link with immune response or host-parasite interaction appears over-interpreted.

R: We are grateful for your suggestion regarding our over-interpretation of the possible host-pathogen interactions mediated by yellow/DCE genes, as well as the possible functional redundancy between DCE and MIF (and D-DT, now corrected at lines 455-457). We better modulated our statements in Discussion and we slightly modified some sentences in Introduction. We agree that melanotic encapsulation is widely (almost only) reported for arthropod, but recent findings suggested the involvement of melanin in the defense responses of one gastropod (Coaglio et al., 2018). Moreover, some examples of dark capsules surrounding pathogens have been reported also for bivalves (see Allam and Raftos, 2015). As you correctly pointed out, since experimental confirmation of the nature of these dark spots is still lacking, it is better to use caution in sustaining the link between DCE/yellow, melanin and innate immunity and to limit our findings to the Mytilus-Mytilicola system (see modifications at lines 461-464; 482-485; 550-556).

For example:

- Introduction section: L35-37: Authors say “The production of a capsule surrounding the invader is followed by its melanization to enclose the invader into an environment with high concentration of ROS [3].” The mounting of a melanized capsule is a well know defense response of arthropods but for most of other invertebrate phyla the encapsulation is not followed by melanization. The link between melanization and immunity is therefore not that clear (or not existing) for most invertebrates and this should be clarified.

- L388-390: Authors observe a higher expression level of DCE in a parasitic copepod as compared with free living copepods and state that “These results suggested that DCE/yellow of M. intestinalis retains a primary role during parasitic interaction and makes combinations of hosts and parasites both possessing DCE/yellow genes interesting cases to study functional gene diversification". Such a general conclusion on the potential importance of DCE would require to observe a higher expression level in several parasitic species.

- In the discussion authors argue that “gene loss might be functionally compensated among DCT, DCE/yellow and D-DT in multiple ways”. This is indeed an interesting idea. However, a potential functional redundancy between DCT or DCE and D-DT remains highly speculative and this should be clarified in the text. Although D-DT (and MIF too, this should be corrected L 453) show a D-dopachrome tautomerase activity, the natural substrate remains unknown and their involvement in melanization has not been functionally demonstrated despite the impressive number of functional studies on these proteins.

Minor points:

- the use of “orthologues” and “paralogues” should be avoided throughout the manuscript as, in most cases, this terminology is not supported by data (e.g. L198, L411, L429)

R: We avoided the unsupported use of these terms along the text and we modified all the other suggested points listed below.

- L102-104: Clarify the fact that the list of selected species is provided in the supplementary data.

- L 150: What are clean reads here? Please, clarify.

R: We modified the sentence to better explain how RNA-seq were used.

-L 156: It may be my mistake but the SRA archive under accession ID PRJNA484693 was not accessible to me.

R: RNA-seq reads are now publicly available (https://www.ncbi.nlm.nih.gov/bioproject/PRJNA484693)

- L441: “combined with multiple, but few HGTs” is somehow contradictory

- HGT: Authors mention the existence of HGT in several instances, but it remains unclear to me what are the evidences of HGT. For example, the positions of the genes -supposed to originate from HGT- in the phylogenetic do not support this statement. Please clarify.

R: We removed this sentence and we modified others to limit possible HGT events to Branchiostoma DCE/yellows, which clustered with the DCE/yellow of the foraminifera Reticulomyxa filosa (lines 440-452).